# The Agricultural–Ecological Benefit of Digital Inclusive Finance Development: Evidence from Straw Burning in China

**Kai Zhao [1,\*], Bintong Yu [1,\*] and Xiaoting Yang [2]**

1   School of Agricultural Economics and Rural Development, Renmin University of China, Beijing 100872, China
2   School of Economics, Beijing Technology & Business University, Beijing 100048, China
*   Correspondence: kaiz@ruc.edu.cn (K.Z.); yubintong@ruc.edu.cn (B.Y.)

**Abstract:** This study provides theoretical and empirical evidence for the agricultural–ecological benefits of digital inclusive finance development. We analyzed the satellite resolution data of agricultural fires and an aggregate development index of digital inclusive finance at the county level in China from 2014 to 2016. The regression analysis demonstrated that digital inclusive finance development can inhibit straw burning, and that the inhibiting effect is more effective in agriculture-oriented counties located in the plain area of the eastern-central developed regions. Additionally, the influence mechanism, whereby digital inclusive finance development may reduce straw burning by improving agricultural mechanization, was also examined. The impact of digital inclusive finance on green agriculture production needs to be explored further since it is a revolutionary mode of financial development.

**Keywords:** digital inclusive finance; green agriculture; straw burning; agricultural mechanization; agricultural modernization

## 1. Introduction

Field straw burning for multiple cropping is a long-standing practice worldwide. Developments in the energy sector and animal husbandry has led to a decline in the use of straw as fuel or livestock feed. Consequently, the phenomenon of straw burning has intensified, especially in developing counties. For example, according to FAO statistics, in China, crop residue burning rose from approximately 58.46 to 68.21 billion tons between 2009 and 2019, which represents 16.65% and 17.48% of global shares, respectively. This is considerably higher than other grain-productive regions, such as the United States of America, European Union, India, and even the aggregate of all South American countries [1]. The amount of straw burned in China, in 2019, is equivalent to approximately 50 billion tons of rural heating coal [2]. This is not only a waste of renewable resources, but also the main source of air pollution.

A large number of studies have confirmed that field straw burning has typical negative economic externalities at both macro and micro levels. Field straw burning releases a series of air pollutants, including PM2.5 (inhalable particles smaller than 2.5 microns), PM10 particles, CO, VOCs (volatile organic compounds), NOx (nitrogen oxides), $SO_2$, PAHs (polycyclic aromatic hydrocarbons), and other carcinogens and toxic compounds, which leads to the deterioration of air quality [3]. Furthermore, the air pollution caused by straw burning has intense temporal and spatial aggregation and heterogeneity, particularly in the harvest season, and the economic and health damage shocks are more damaging than that caused by industrial pollution [4]. Particularly, biomass burning and its haze pollution causes respiratory ailments, cardiovascular diseases, strokes, and several types of cancers, and leads to premature deaths [5]. Additionally, it induces depression [6], cognitive decline [7], decreases in labor productivity [8], and even disrupts people's capacity for investment and economic decisions [9–11]. These adverse physical and mental health

effects are bound to inevitably lead to an increase in medical expenditure [12], migration costs [13], and pollution prevention costs, through the purchase of equipment, such as anti-smog face masks and air cleaners [14].

Considering the multi-dimensional hazards of biomass burning and haze pollution, several studies have analyzed the causes of field burning and proposed countermeasures. According to Lin and Begho [15], the major reason for agricultural fire is the relatively high temporal and economic costs of harvesting, transporting, and displacing straw compared to burning it. The shortening of the time window between harvesting and cropping through developments in multiple cropping and tillage systems, and the increase in farm labor cost caused by decreasing rural populations on account of urbanization has greatly contributed to this. Additionally, farmers' environmental consciousness and 'peer effects' also impact field burning behavior [16,17]. Therefore, administrative environmental regulation interventions, including pollution punishment and environmental protection incentives are necessary [16,18]. However, several studies argue that the impact of environmental regulation on field burning behaviors is limited [19,20]; thus, regulations alone cannot address this issue.

Many studies observed that mechanization of cropping can make straw residue processing and transformation cost-effective [15,16,21,22]. However, agricultural machines are a relatively large capital investment for farmers, who face tight budget constraints; similarly, leasing agricultural machinery or purchasing mechanical harvesting services can be unaffordable to farmers prior to harvesting [23]. Therefore, easing budget constraints and relieving temporary savings shortages among farmers is essential for alleviating straw burning. Although government-supported agricultural mechanization projects and subsidies can theoretically make progress, the impact is heterogeneous and not always efficient [24]. Meanwhile, obtaining loans from conventional financial institutions is difficult for farmers because of information asymmetry and high transactional costs [25].

Digital inclusive finance might be a modern solution to this old problem. Digital inclusive finance has two parts: inclusive finance and digital finance. Inclusive finance, also known as financial inclusion, is defined as a low-cost and efficient financial system characterized by comprehensiveness, multi-levelness, and popularity, especially in providing convenient financial services for low-income groups in less developed areas [26,27]. Meanwhile, digital finance, also known as online finance or internet finance, is defined as an emerging financial operation mode for financing, payment, investment, and information intermediary services using internet technology and information communication technology [28–30]. Compared to conventional financial services, digital inclusive finance can broaden financial accessibility, promote information symmetry, strengthen financial risk prevention and control, and reduce bad debt rate. Essentially, digital inclusive finance involves the use of digital technology to offer financial services to individuals and small businesses traditionally underserved by traditional financial institutions. In addition, digital inclusive finance seeks to improve the efficiency and effectiveness of financial services for low-income and unbanked populations, as well as to increase access to financial services for these populations [31].

Digital inclusive finance's history can be traced back to the early 2000s, when mobile phone technology was used for financial transactions in developing countries. This was followed by the emergence of digital financial services providers, such as M-PESA in Kenya, which uses mobile money to provide basic financial services to the unbanked population [32]. The rapid development of digital infrastructure and mobile communication technology in recent years has led to explosive growth in digital inclusive finance. According to the World Bank, the number of mobile money accounts in developing countries increased from 2 million in 2008 to over 800 million in 2018. Furthermore, the number of digital transactions in these countries has also been growing, reaching over 1 trillion USD in 2018 [33]. Moreover, the transactional volume of digital inclusive finance in China reached approximately CNY 40 trillion in 2020, increasing by roughly 5% annually [34]. Moreover, in rural areas, in contrast with the typical household loan of CNY 100 thousand from a local

bank, the digital inclusive credit service is more minor, ranging from CNY 20 thousand to 50 thousand n, which covers two thirds households in need of microcredit [35]. This indicates that digital inclusive finance offers supplementary services to small and sinking rural households.

Research conducted on the impact of digital inclusive finance on agricultural production and rural development in developing countries in Africa and Asia has revealed that digital inclusive finance improves the accessibility of financial services to small farmers and establishes a more inclusive rural credit system, which is conducive to the promotion of large-scale operations, the increase of agricultural productivity, and agricultural modernization [29,36,37]. Additionally, studies have found that the development of digital inclusive finance plays a positive role in promoting the scientific and technological innovation of agricultural enterprises and the entrepreneurial decisions of rural residents [38,39]. Furthermore, digital inclusive finance development positively impacts ecological progressiveness as it increases green economic efficiency, promotes green innovation, and improves air quality [40–42].

Recently, scholars have increasingly drawn attention to the importance of the ecological impacts of digital inclusive finance development, particularly in the agricultural sector and rural areas of the country. However, taking into account the characteristics of digital inclusive finance, scholars also have three different views on this issue. In the beginning, a majority of studies have demonstrated that the development of digital inclusive finance can effectively promote the development of agriculture as a whole. Hong et al. found that China's agricultural green total factor productivity can be significantly increased through financial inclusion, owing to the optimization of the agricultural industry structure with provincial level data [43]. Secondly, a portion of scholars argues that digital inclusive finance development is still in the early stage and there has been a relatively slow development of digital infrastructure in rural areas when compared with urban areas, where there is still a great deal of difficulty for the agricultural sector and rural areas to access financial resources and benefit from financial development [26,30,44]. Moreover, empirical research found that the ecological impact of digital inclusive finance is heterogeneous. Guo et al. found that agricultural green development is affected differently in different regions over the short and long term by digital inclusive finance [45]. Accordingly, the ecological impact of financial development remains controversial regardless of the experience of other countries or the current state of China.

Additionally, based on the existing literature, the influence of digital inclusive finance development on straw burning, which is one of the most immediate agricultural and rural ecological threats, can be categorized into two categories: direct effect and indirect effect. On the one hand, the development of digital inclusive finance has direct impacts on poverty reduction as well as easing budget constraints. People in less developed and rural areas can acquire assets and build personal credit through digital inclusive finance, so they can invest in relatively advanced methods of agriculture production, such as buying a harvester or hiring a mechanical harvesting service [36,46]. As a result, they are able to reduce straw burning while saving labor costs at the same time [15,22]. On the other hand, the development of digital inclusive finance has indirect impacts on region economic growth promoting and environmental needs growing. According to the Kurnitz curve hypothesis applied to environmental quality, as regional and individual economic conditions improve, the demand for higher levels of quality will also rise in proportion to those improvements [47]. Thus, in order to reduce pollution caused by burning straw, farmers are actively and passively changing the way straw is treated. Harvesting with machinery is always the most widespread method of non-burning straw treatment [16]. There are, however, some limitations to the potential impact of digital inclusive finance on straw burning. Accessibility and effectiveness of digital inclusive finance are affected by the availability and quality of digital infrastructure, such as internet connectivity and mobile networks [26]. Furthermore, cultural and social factors, such as traditional practices and attitudes towards technology, can also influence the adoption and impact of digital inclusive

finance [45]. Although several studies discussed the combined effect of digital inclusive finance, there are still no studies that verify the influence of digital inclusive finance development on straw burning through mathematical models and empirical analysis. Though various studies have discussed the effects of digital inclusive finance, there are still no studies directly verifying the influence of digital inclusive finance development on straw burning by using mathematical models or empirical tests, which inspired this study.

Having reviewed the above analysis, it is clear that there are a number of shortcomings with the previous research that has been conducted. On the one hand, rather than focusing on a specific mode of farming pollution production, the present studies focus on green total factor productivity. It is sensitive to measurement methods and variable selection. On the other hand, related empirical research mainly use provincial level data with small sample size and large statistical noise. Moreover, to analyze the theoretical framework, the current study relied on text-based logical descriptions rather than mathematical models. Based on this, this paper makes three major contributions to the existing research in the domain. First, this is the first study to analyze the agriculture ecological benefit of digital inclusive finance development on a theoretical and empirical basis. We provide evidence from China to support the argument that digital inclusive finance development is beneficial for agriculture from an ecological standpoint. Next, we focus on the difference caused by geographic and spatial heterogeneity in the endowment of agricultural resources. Furthermore, based on previous studies and theoretical model derivation, we explore the possible influence mechanism of digital inclusive finance development on reducing straw burning by improving agricultural mechanization, and utilize county-level panel data in China to verify whether the mechanism is established. Thus, our analysis contributes to the growing literature on understanding the ecological benefit of finance development from a new perspective, particularly in developing countries.

The organization of the paper is as follows. Section 2 contains the theoretical analysis. Section 3 presents the data and methodology. The empirical analysis is presented in Section 4. Section 5 presents the conclusion.

## 2. Theoretical Analysis

The theoretical framework was developed by expanding a standard household model by adding the financial sector in a two-period model. It is a simplified partial equilibrium model focused on only the financial market clearing. The markets are assumed to be complete, and the prices are assumed to be exogenous. It is assumed that a representative agricultural household maximizes utility and allocates their agricultural income optimally between consumption and farm investment. Additionally, it is assumed that in period one, the household borrows from the financial sector to purchase factors of production for crop planting and consumption; in period two, the household disposes of the straw residues (burning or recycling), harvests the crop, and then sells it to the market for repaying loans to the financial sector and for consumer activities. In this context, the model can be written as follows:

$$max\ U(C_1, C_2) = max\ [U(C_1) + U(C_2)] \tag{1}$$

$$s.t.\ C_1 + K \leq m \tag{2}$$

$$C_2 + m(1 + R) + k(1 - \alpha)\varphi Q \leq PQ \tag{3}$$

where $C_1$ and $C_2$ represent the household's consumption activities in two periods to maximize utility. The income constraint, $K$, represents the cost of planting production factors excluding investment in agricultural machinery on crop straw residues processing. $m$ and $R$ represent the quantity of loans and the loan interest rate, respectively. $k$ represents the investment in agricultural machinery on crop straw residue processing, $Q$ represents the quantity of crop output, $P$ represents the price of crop output, and $\varphi$ represents the coefficient of the amount of straw that can be generated by one unit crop output. Meanwhile, $\alpha$ represents the ratio of straw burning and $1 - \alpha$ represents the ratio of straw recycling, such as mechanized straw binding and mulching.

Then, we construct the Lagrange equation and take derivatives for $C_1$, $C_2$, $K$, $m$:

$$L = U(C_1) + U(C_2) + \lambda_1[m - C_1 - K] + \lambda_2[PQ - C_2 - m(1 + R) - k(1 - \alpha)\varphi Q] \quad (4)$$

$$\frac{\partial L}{\partial K} = -\lambda_1 + \lambda_2 \left[ P \cdot \frac{\partial Q}{\partial K} - k(1 - \alpha)\varphi \cdot \frac{\partial Q}{\partial K} \right] = 0 \quad (5)$$

$$\frac{\partial L}{\partial m} = \lambda_1 - \lambda_2(1 + R) = 0 \quad (6)$$

On the other hand, the financial sector maximizes profit and the equation can be written as follows:

$$max\ \pi = [R - (1 - \delta)h + 1]M \quad (7)$$

where $\delta$ represents the digital inclusive finance coefficient ($0 < \delta < 1$), $h$ represents a coefficient of shadow cost for bad debt, and $M$ represents the quantity of lending loans. The development of digital inclusive finance, $\delta$, can improve information symmetry in the financial market and increase the accessibility of credit for the borrower to reduce the shadow cost of bad debt of the financial sector [28–30,48]. Thus, the actual bad debt shadow cost under digital financial inclusion intervention becomes the term of $(1 - \delta)h$.

When the household and financial sector achieve equilibrium and the financial market clears, the quantity of loans lent by the financial sector equals the loans borrowed by the household, as $m = M$. The derivatives for $m$ are thus:

$$\frac{\partial \pi}{\partial M} = R - (1 - \delta)h = 0 \quad (8)$$

Based on Equations (5), (6), and (8), we get Equations (9) and (10) with respect to $\alpha$ and $k$:

$$\alpha = \frac{u[1 + (1 - \delta)h]}{k\varphi \cdot \frac{\partial Q}{\partial K}} + \frac{k\varphi - p}{k\varphi} \quad (9)$$

$$k = \frac{p - u[1 + (1 - \delta)h]}{(1 - \alpha)\varphi \cdot \frac{\partial Q}{\partial K}} \quad (10)$$

Then, we take derivatives for $\delta$ in Equations (9) and (10):

$$\frac{\partial \alpha}{\partial \delta} = -\frac{hu}{k\varphi \cdot \frac{\partial Q}{\partial K}} < 0 \quad (11)$$

$$\frac{\partial k}{\partial \delta} = \frac{hu}{(1 - \alpha)\varphi \cdot \frac{\partial Q}{\partial K}} > 0 \quad (12)$$

In practice, finance is still an essential component of digital inclusive finance. According to Equation (11), under digital financial inclusion intervention, credit availability on the market is generally increasing. As a result, financing and liquidity constraints are alleviated, regional credit resources are allocated better, and the modernization and innovation of green agricultural production are improved [49]. In other words, by increasing the distribution of inclusive financing, farmers are able to loosen their budget constraints, and therefore they will be able to increase their incomes from agriculture. Meanwhile, a change in production mode and environmental demand resulted in farmers reducing the proportion of straw burned, based on the income effect [29,47]. Thus, it demonstrates that the development of digital inclusive finance has a negative correlation with the straw burning behavior of the household. In light of this, research Hypothesis 1 is proposed in this paper:

**Hypothesis 1.** *A significant reduction in straw burning can be achieved through the development of digital inclusive finance.*

Similarly, along with the development of digital financial inclusion, all services can be operated online. Financial services are available to remote rural households in this way, alleviating the insufficient supply of monetary resources caused by geographic factors [46]. According to Equation (12), inputs for green agricultural production can be promoted by easy credit to farmers. Specifically, in order to dispose of straw and avoid burning, farmers are more likely to buy a harvester or to hire a mechanical harvesting service because of the softening of budgetary restraints [21]. It greatly reduces the time and costs associated with straw non-burning processes, such as straw turnover and recycling [19], because straw burning will be reduced in the region as a result of the increasing demand for agricultural mechanization.

**Hypothesis 2.** *The development of digital inclusive finance promotes agricultural machinery for straw residue processing and recycling, which is an effective and efficient path for reducing straw burning.*

In general, as a crucial part of the study, the theoretical framework analysis was carried out as shown in Figure 1.

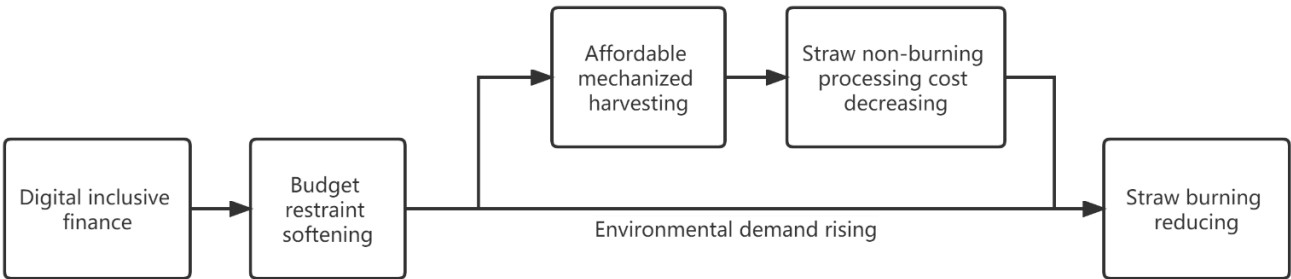

**Figure 1.** Theoretical framework analysis diagram.

### 3. Data and Methodology

#### 3.1. Data

Empirical analysis was conducted based on county-level data in China from 2014 to 2016 to examine the impact of digital inclusive finance on green agriculture production. In selecting this period for analysis, three main reasons were considered. To begin with, the Digital Inclusive Finance Index for counties was published for the first time in 2014. Secondly, a revision of the Law on Prevention and Control of Air Pollution was introduced at the end of 2016 on a national scale that recognized straw burning as an important source of air pollution. Our focus was on the period before the law took effect, since environmental regulation could seriously interfere with the agricultural-ecological benefits of digitally inclusive finance development. Furthermore, the algorithm method for identifying fire points of crop residue burning changed after 2016, so we selected data before 2017.

The explanatory variable, green agriculture production, is measured by the number of straw-burning fires (denoted by *Fire* in regression). NASA's moderate resolution imaging spectroradiometers (MODIS) are capable of detecting crop residue burning through remote sensing. The detection of fire points is based on the detection of an anomaly within a pixel (50 square meters) using a contextual algorithm that exploits mid-infrared radiation from the fire. Straw burning can be distinguished from other types of fires based on land use and geography information; thus, it is consistent and can be compared across the year and between counties [50]. Additionally, since the peak fire season occurs between October and November after the autumn harvest, we collect the data regarding fire points in those months (*Autumnfire*) for the robustness test. Burnt area estimation is not feasible for the study because of large uncertainties in the algorithm.

The primary explanatory variable Is digital inclusive finance, measured by an aggregate index (*DIFindex*) from the China Digital Inclusive Finance Development Index Report

released by the Digital Finance Research Center of Peking University. The Alibaba Group's Alipay financial account big data is used to create the index; thus, it is based on actual information for each transaction, which is representative and reliable [34,48]. Additionally, it is the only data that can be considered as the proxy variable of financial inclusion and digital finance developmental level in China. Furthermore, another crucial explanatory variable is agricultural mechanization, measured by the total power of agricultural machinery (*Machine*) in million kilowatt-hours, from the County statistical yearbook.

The following are the control variables used in this study: the annual average temperatures (*Temp*) in degrees Celsius and the annual cumulative precipitation (*Precip*) in millimeter are reported by the National Meteorological Information Center. Temperature and rainfall play a substantial role in agriculture and straw burning as proxy variables for local climate. Haze pollution measured by the Atmospheric Composition Analysis Group at Washington University in St. Louis was measured using a mean annual concentration of fine particulate matter (*PM25*, particles smaller than 2.5 microns). The heavier the local air pollution, the lower the intensity of local environmental regulations, and therefore, the more willing farmers are to burn straw burn. The density of population (*Popd*) in person/km$^2$; the fiscal expenditure (*Fexpend*) in ten thousand CNY; the total grain output (*Grain*) in ton; and the proportion of primary industry employees of total employed population (*Agriworker*) obtained from the County statistical yearbook. Local economic and agricultural characteristics are controlled by these four variables.

The data sources used for the study are presented below, and the summary statistics for the variables used in the analysis are shown in Table 1.

**Table 1.** Descriptive statistics.

| Variables | Obs. | Mean | Std. Dev. | Min | Max |
|-----------|------|------|-----------|-----|-----|
| *Fire* | 6280 | 56.1 | 124.6 | 0 | 1966 |
| *Autumnfire* | 6280 | 12.62 | 65.37 | 0 | 1328 |
| *DIFindex* | 6283 | 71.83 | 22.15 | 10.24 | 131.8 |
| *Machine* | 5414 | 47.05 | 42.42 | 1 | 336 |
| *Temp* | 5927 | 1064 | 529.1 | 0 | 2785 |
| *Precip* | 5921 | 14.12 | 5.115 | −2.826 | 25.17 |
| *PM25* | 5927 | 44.22 | 18.64 | 1.222 | 139.2 |
| *Popd* | 5899 | 662.4 | 2367 | 0.155 | 43,946 |
| *Fexpend* | 5464 | 301,300 | 203,861 | 15902 | 2,690,000 |
| *Grain* | 5041 | 281,394 | 303,036 | 12 | 3,393,000 |
| *Agriworker* | 5350 | 0.728 | 0.14 | 0.00138 | 0.976 |

### 3.2. Methodology

The following empirical equation was developed to test the hypothesis:

$$Y_{it} = \beta_0 + \beta_1 X_{it} + \beta_2 \text{Control}_{it} + \mu_t + \rho_i + \varepsilon_{it} \tag{13}$$

where $Y_{it}$, $X_{it}$, and $\text{Control}_{it}$ represent the number of straw fires, the Digital Inclusive Finance Index, and the control variables of local weather and agricultural–economic conditions of the i-th city in the t-th year, respectively. Additionally, $\mu_t$, $\rho_i$, and $\varepsilon_{it}$ represent the time fixed effect, entity fixed effect, and the error term, respectively.

## 4. Empirical Results

### 4.1. Baseline Model and Endogeneity Issue

As a result of the gradual addition of control variables in columns (1)–(3), Table 2 displays the impact of digital inclusive finance on the number of fires caused by straw burning. We found that the *DIFindex* coefficient is always negative and it is statistically significant at the 1% level. It indicates that the growth of digital inclusive finance can effectively inhibit straw burning behavior, and reduces the accompanying environmental pollution and biomass resource waste. It is consistent with the theoretical analysis and

existing research about the ecological progressiveness effect of the digital inclusive finance development [39–41]. Additionally, the result supports the premise that digital inclusive finance promotes sustainable agricultural practices (SAPs) development [51]. It should be noted that a certain proportion of counties have no fires or occasionally burn with a few fires, so the sample of our study may not be representative at the national level by taking the logarithm of the key-dependent variable *Fire*. Thus, we used the penal passion model for the robustness test. Moreover, in this study, robust standard errors clustered at the county level were used to solve the issues of heteroscedasticity and intra-class correlation, which are frequently used in econometric regressions to avoid biased estimates [52,53].

**Table 2.** Basic regression and endogeneity.

| | **(1) FE-Panel** | **(2) FE-Panel** | **(3) FE-Panel** | **(4) IV-2SLS** |
|---|---|---|---|---|
| **Dependent Variables** | *Fire* | *Fire* | *Fire* | **IV (*Distance to Hangzhou*)** *Fire* |
| *DIFindex* | −0.508 *** (0.1068) | −0.437 *** (0.1039) | −0.498 *** (0.1107) | −2.593 *** (0.8702) |
| *Precip* | | −0.040 *** (0.0070) | −0.046 *** (0.0058) | −0.066 *** (0.0095) |
| *Temp* | | 28.277 *** (5.3012) | 22.068 *** (4.6170) | 13.930 *** (4.3052) |
| *PM25* | | 1.369 *** (0.2677) | 1.124 *** (0.2809) | 1.029 *** (0.2798) |
| *Popd* | | | 0.440 *** (0.1507) | −0.413 (0.3756) |
| *ln_Fexpend* | | | 17.018 * (9.6778) | 23.905 ** (10.8156) |
| *ln_Grain* | | | 8.783 * (4.6587) | 17.093 ** (6.7578) |
| *ln_Agriworker* | | | −5.496 (7.4757) | 0.180 (9.1137) |
| _cons | 96.391 *** (5.9361) | −330.212 *** (83.0945) | −597.570 *** (209.0603) | −379.95 (271.11) |
| Time and County FE | Yes | Yes | Yes | Yes |
| N | 6280 | 5918 | 4659 | 4419 |
| adj. R-sq | 0.055 | 0.098 | 0.122 | 0.035 |
| F | 47.879 | 43.751 | 27.198 | 26.986 |

Note: * $p < 0.1$, ** $p < 0.05$, *** $p < 0.01$. The robust standard errors clustered at the county level in parentheses.

Moreover, because of the endogeneity issues between digital inclusive finance development and agricultural pollution behaviors, the interaction term of the distance from each county to the headquarters of Alipay in Hangzhou and year dummy variables as an instrumental variable (IV) with the two-stage least square (2SLS) method were used to eliminate endogeneity. The data source of *DIFindex* mainly comes from Alipay, and its headquarters is located in Hangzhou. Thus, the distance to Hangzhou represents the development level of digital inclusive finance along with the business expansion of Alipay [48]. Additionally, the geographical distance to Hangzhou is exogenous for straw burning activities, so it is a valid instrumental variable. According to the result in column (4) of Table 2, digital inclusive finance development still has a significant negative impact on regional straw burning activities. This demonstrates that the estimated coefficients using the IV model are larger compared with the FE-Panel model in column (3) caused by the measurement error of independent variables [54–56]. Specifically, the farmlands with straw burning fires are usually concentrated in the non-central and underdeveloped area (financially and digitally) inside a county. Therefore, its digital inclusive finance level should be lower than the average county-level of *DIFindex*, and the coefficients of FE-Panel model are underestimated.

Table 3 demonstrates that according to the first stage of 2SLS regression results, the *t*-test and F-test values, under-identification test, and weak identification test are statistically significant, thus, the instrumental variable is valid.

**Table 3.** Summary results for first-stage regressions of IV-2SLS.

| Dependent Variable: *Distance to Hangzhou* | Time and County Fixed Effects Estimation with Control Variables | | | | Under-Identification Test | | Weak Identification Test | |
|---|---|---|---|---|---|---|---|---|
| | Coef. | t | F(1, 1484) | *p*-Value | Stock-Yogo Chi-sq (1) | *p*-Value | Stock-Yogo F(1, 1484) | 10% Maximal IV Size |
| *Fire* | 0.304 *** | 8.57 | 73.42 *** | 0.00 | 73.62 *** | 0.00 | 73.42 *** | 16.38 |

Note: *** $p < 0.01$. Statistics robust to heteroskedasticity and clustering on county level.

### 4.2. Robustness Test

The autumn fire (a fire peak season that occurs between October and November after the autumn harvest) was used as the dependent variable alternative in columns (1) and (2) of Table 4 to test the robustness of our findings. Then the panel-Poisson model in column (3) was used to test the impact of a certain proportion of the sample with no fires. Finally, the sample was winsorized at the 1th and 99th percentiles in columns (4), and the dataset was transferred into the balanced panel data in columns (5). The regression results demonstrate that the inhibition effect of digital inclusive finance development on straw burning activities is robust.

**Table 4.** Robustness test.

| | (1) FE-Panel | (2) IV-2SLS | (3) Poisson-Panel | (4) IV-2SLS Winsorizing (1,99) | (5) IV-2SLS Balanced-Panel |
|---|---|---|---|---|---|
| **Dependent Variables** | *Autumnfire* | *Autumnfire* | *Fire* | *Fire* | *Fire* |
| *DIFindex* | −0.311 *** | −2.488 *** | −0.007 *** | −1.624 *** | −2.634 *** |
| | (0.0940) | (0.6357) | (0.0004) | (0.6124) | (0.8813) |
| Controls | Yes | Yes | Yes | Yes | Yes |
| Time and County FE | Yes | Yes | Yes | Yes | Yes |
| N | 4423 | 4413 | 4396 | 4301 | 4413 |
| R-sq | 0.061 | 0.003 | | 0.114 | 0.031 |
| F | 6.856 | 4.065 | | 32.654 | 26.912 |
| Wald chi2 | | | 13,815.95 | | |

Note: *** $p < 0.01$. The robust standard errors clustered at the county level in parentheses.

### 4.3. Heterogeneity Analysis

The environmental effect of digital inclusive finance development may differ depending on the geographical location in which the counties are located. The interaction terms *DIFindex* ∗ *plain*, *DIFindex* ∗ *agri*, and *DIFindex* ∗ *eastmid* were added to the regression to examine whether there is heterogeneity at the county level. Specifically, *flat* represents the dummy variable of plain areas, and when the gradient of the county is below the national average level (12 degrees), it is taken as 1, and it is taken as 0 for non-flat areas. Similarly, *agri* represents the dummy variable of the agriculture-oriented county with the proportion of arable land above 70%, and *eastmid* represents the county located in the eastern-central developed regions. The results shown in Table 5 demonstrate that geographic differences affect the relationship between digital inclusive finance development and straw burning activities. That is, the environmental effect of digital inclusive finance development is greater in the agriculture-oriented county located in the plain area of the eastern-central developed regions. This can be explained by the higher demand for clean air among people in the developed regions in the agriculture-oriented plain counties, which are also the fire-ridden areas [47]. Thus, digital inclusive finance development can be considered an efficient path for reducing agricultural pollution.

**Table 5.** Heterogeneity.

| | **(1) FE-Panel** | **(2) FE-Panel** | **(3) FE-Panel** |
|---|---|---|---|
| **Dependent Variables** | *Fire* | *Fire* | *Fire* |
| *DIFindex* | −0.436 *** | −0.471 *** | −0.407 *** |
| | (0.1003) | (0.1026) | (0.0978) |
| *DIFindex ∗ plain* | −0.338 *** | | |
| | (0.1217) | | |
| *DIFindex ∗ agri* | | −0.346 * | |
| | | (0.1769) | |
| *DIFindex ∗ eastmid* | | | −0.222 ** |
| | | | (0.1013) |
| Controls | Yes | Yes | Yes |
| Time and County FE | Yes | Yes | Yes |
| N | 4659 | 4659 | 4659 |
| adj. R-sq | 0.127 | 0.126 | 0.124 |
| F | 30.606 | 31.507 | 26.947 |

Note: * $p < 0.1$, ** $p < 0.05$, *** $p < 0.01$. The robust standard errors clustered at the county level are presented within parentheses.

### 4.4. Mechanism

Investigating the mechanism through which digital inclusive finance affects straw burning is of great significance in clarifying the relationship between these two variables. The analysis of the theoretical model demonstrates that agricultural mechanization can also be promoted through digital inclusive finance. The study used the total power of agricultural machinery (calculated by natural logarithm) as a proxy for agricultural mechanization and analyzed the mechanism. Based on the results presented in columns (1) of Table 6, developing digital inclusive finance can significantly improve agricultural modernization. Meanwhile, agricultural modernization development can effectively inhibit straw burning activities, as shown in columns (2)–(4). Hence, digital inclusive finance development can reduce straw burning activities through the channel of stimulating agricultural mechanization, which is consistent with our theoretical analysis and previous studies [29,36,37].

**Table 6.** Mechanism of stimulating agricultural mechanization.

| | **(1) FE-Panel** | **(2) FE-Panel** | **(3) FE-Panel** | **(4) XTPoisson** |
|---|---|---|---|---|
| **Dependent Variables** | *Ln_Machine* | *Fire* | *Autumnfire* | *Fire* |
| *DIFindex* | 0.002 *** | | | |
| | (0.0004) | | | |
| *Ln_Machine* | | −25.908 *** | −14.348 *** | −0.339 *** |
| | | (5.3445) | (4.4412) | (0.0229) |
| Controls | Yes | Yes | Yes | Yes |
| Time and County FE | Yes | Yes | Yes | Yes |
| N | 4844 | 4483 | 4483 | 4464 |
| R-sq | 0.096 | 0.124 | 0.067 | |
| F | 53.373 | 29.126 | 7.214 | |
| Wald chi2 | | | | 14,218.39 |

Note: *** $p < 0.01$. The robust standard errors clustered at the county level in parentheses.

*4.5. Empirical Results Discussion*

Based on the empirical results of basic regression, it indicates straw burning can be effectively reduced through digital inclusive finance. The quantitative model and theoretical analysis support these results. Furthermore, it provides new theoretical evidence for the debate of the relationship of digital inclusive finance development and ecological progressiveness [41,43]. Moreover, in heterogeneity analysis, the agriculture ecological benefit of digital inclusive finance development is more effective in a relatively developed area. It indicates that ecological benefit is heterogeneous [48]. Meanwhile, the inclusive feature of digital inclusive finance is still not entirely reflective, which agrees Several studies question the effect of digital inclusive finance development [46]. In addition, the mechanism investigation finds that digital inclusive finance development leads to a reduction in straw burning activities by enhancing agricultural mechanization. It validates existing research that farmers generally process straw in a non-incinerating manner by employing economical agricultural mechanization after budget constraints are relieved [17].

## 5. Conclusions

Based on the satellite resolution data of agriculture fire and an aggregate index of digital inclusive finance at the county level in China from 2014 to 2016, this study discusses the agriculture ecological benefit of digital inclusive finance development theoretically and empirically, using field straw burning as a proxy. The study utilized the 2SLS model to address the endogeneity problem associated with agriculture fire by using the distance between Hangzhou and each county as the instrument variable. The study demonstrates that digital inclusive finance development can inhibit straw burning, especially in the autumn harvest season, after controlling for climate and socio-economic characteristics. Additionally, the results demonstrate that the inhibiting effect is more severe in agriculture-oriented counties located in the plain area of the eastern-central developed regions. Furthermore, the study examined the influence mechanism, whereby, digital inclusive finance development may reduce straw burning by improving agricultural mechanization.

It is suggested that the following policy implications be put forward based on the conclusions. First of all, developing a digital inclusive finance ecosystem in rural areas is important. There is strong evidence that digital inclusive finance can facilitate agricultural mechanization and reduce straw burning. Governments should provide legal and financial assistance to financial institutions to ensure the supply and quality of financial services in rural areas. Secondly, enhancing rural digital infrastructure and monetary facilities is also a priority. There is a limitation to the availability of digital inclusive finance owing to the lack of digital financial infrastructure. Because of this, rural digital financial infrastructure should be enthusiastically promoted by the government. Last but not least, subsidizing farm machinery through the digital inclusive finance system is another important point to consider, as is developing localized digital inclusive financial policies. Various development policies should be adopted to improve agricultural sustainability production, depending on the characteristics of development and financial resources in different regions.

The findings have substantial policy implications for promoting rural and agricultural ecological progressiveness, particularly in developing countries. Promoting the development of digital inclusive finance has considerable positive effects on agricultural modernization and rural ecological progress in countries with relatively underdeveloped traditional financial markets such as China. Furthermore, the development of digital inclusive finance in mountainous areas and less-developed western regions should receive more attention from governments and enterprises engaged in digital financial cooperation.

However, this study has two major limitations. First, the satellite resolution data of agricultural fire can reflect the degree of straw burning in a county, but it is not entirely reflective of farmers' straw burning behavior. Second, this study is focused exclusively on straw burn, without considering other polluting activities, such as the excessive use of pesticide and fertilizer. Given the increasing concerns regarding the rural and agricultural eco-environment, further research on other major pollution behaviors may help in improv-

ing our understanding of the effect and mechanism of digital inclusive finance in a broader context, especially by using farmer household micro-data.

**Author Contributions:** Conceptualization, K.Z., B.Y. and X.Y.; methodology, K.Z., B.Y. and X.Y.; formal analysis, K.Z., B.Y. and X.Y.; data curation, K.Z.; writing—original draft preparation, K.Z.; writing—review and editing, K.Z.; supervision, K.Z. All authors have read and agreed to the published version of the manuscript.

**Funding:** This research received no external funding.

**Institutional Review Board Statement:** Not applicable.

**Informed Consent Statement:** Informed consent was obtained from all subjects involved in the study.

**Data Availability Statement:** Data are available from the authors upon reasonable request.

**Conflicts of Interest:** The authors declare no conflict of interest.

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
