# Peer review of "The Agricultural–Ecological Benefit of Digital Inclusive Finance Development: Evidence from Straw Burning in China"

_sustainability, doi:10.3390/su15043242_

Round 1
Reviewer 1 Report
The argument is well-structured and the introduction is strong. The work makes clear both the theoretical and practical contributions. The paper might benefit from some modifications. The coverage of inclusive digital finance in rural areas China needs to be supported by data because the reference is DFI in urban areas. Descriptive statistics from Table 1 should be in the Empirical Results section. The results needs to be discussed by elaborating the current state of China's practices of straw burning and the determinants as well as by relating the findings to those of the previous studies. Please check the references' consistency in writing.

Author Response
Dear Reviewer 1,
Thank you for your useful insights and suggestions, which have enriched the manuscript and produced a better and more balanced account of the research. We have modified the manuscript accordingly, and detailed corrections are listed below:
- How big is the coverage of digital inclusive finance in rural areas of China? Because the article discussed about DFI and urban innovation.
Response: We concur and hence have included more information to reflect the coverage of digital inclusive finance in rural areas of China:
“Moreover, in rural areas, in contrast with the typical household loan of 100 thousand yuan from a local bank, the digital inclusive credit service is more minor, ranging from 20 thousand to 50 thousand yuan, which covers two thirds households in need of micro-credit [35]. This indicates that digital inclusive finance offers supplementary services to small and sinking rural households.”
- Why is this period selected for analysis?
Response: In selecting this period for analysis, three main reasons were considered. To begin with, the Digital Inclusive Finance Index for counties was published for the first time in 2014. Secondly, a revision of the Law on Prevention and Control of Air Pollution was introduced at the end of 2016 on a national scale that recognized straw burning as an important source of air pollution. Our focus was on the period before the law takes effect since environmental regulation could seriously interfere with the agricultural-ecological benefits of digitally inclusive finance development. Furthermore, the algorithm method for identify-ing fire points of crop residue burning changed after 2016, so we selected data before 2017.
We add this part in our manuscript for explanation.
- I think this table should move to Empirical Results.
Response: In social science research report format, the table of descriptive statistics is usually followed by data introduction. We still move the table to the end of section in order to be close to empirical results.
- Please extend the title of this table
Response: We extend the title of this table as “Mechanism of stimulating agricultural mechanization.”
- Before Conclusions, please discuss the results by communicating this study's findings with those of previous studies.
Response: We concur and hence have included more information to discuss the results by communicating this study's findings with those of previous studies.:
“4.5. Empirical Results Discussion
Based on the empirical results of basic regression, it indicates straw burning can be effectively reduced through digital inclusive finance. The quantitative model and theoretical analysis support these results. Also, it provides new theoretical evidence for the de-bate of the relationship of digital inclusive finance development and ecological progressiveness [41,43]. Moreover, in heterogeneity analysis, the agriculture ecological benefit of digital inclusive finance development is more effective in relatively developed area. It in-dictates that ecological benefit is heterogeneous [48]. Meanwhile, the inclusive feature of digital inclusive finance is still not entirely reflective, which agrees Several studies ques-ton the effect of digital inclusive finance development [46]. Besides, the mechanism investigation finds that digital inclusive finance development leads to a reduction in straw burning activities by enhancing agricultural mechanization. It validates existing research that farmers generally process straw in a non-incinerating manner by employing eco-nomically agricultural mechanization after budget constraints are relieved [17].”
- Title of article should be lower case, pls check other articles
Response: Thank you for your suggestion. We double check the format of References and make some modifications.
We look forward to hearing from you and would be happy to make further changes, if required.
Reviewer 2 Report
The author presents "The Agricultural-Ecological Benefit of Digital Inclusive Finance Development: Evidence from Straw Burning in China". This manuscript has some relevance, but also has the following problems:
1. Introduction part: The introduction of digital inclusive financial background and development status is insufficient; the connection between digital inclusive finance and agricultural straw incineration is not very close.
2. The sorting out of the current status of the existing literature is not detailed and clear, and the literature review part is a bit thin.
3. Putting the hypothesis that the demonstration there is not clear enough. The article draws two assumptions through formulas, and it is recommended to add some text expressions.
4. The study time period is short, and it is recommended to expand the inspection time.
5. The description of the variable is too brief, it is recommended to explain the variables in the permanent formula in detail.
6. The mechanism analysis is not sufficient enough. The empirical results support the path of agricultural mechanization but the relevant theoretical expression is a bit thin, and there is no in-depth explanation and analysis.
7. Robustness test is a bit less. The research results are credible discounts. It is recommended to increase the robustness test.
8. Policy suggestions are not closely linked to the research conclusions, and it is recommended to modify it.
9. There are some minor grammatical problems, such as“the and summary statistics for
the variables”.It is recommended that the author check the manuscript.
Reviewer 3 Report
An interesting study, however, requiring appropriate correction. Major comments 1. The authors should theoretically demonstrate a clear link between straw burning and financial digitization. This relationship has not been properly explained, and one cannot proceed directly to the econometric analysis without indicating such a relationship and its reasons. 2. What is the research gap that the manuscript fills? This requires clarification on the basis of the literature analysis carried out, and then an indication of the contribution to the current state of the literature. 3. How was the problem of heteroskedasticity checked in the model? What tests have been done in this regard? 4. What is the main goal of the conducted research? This must be clearly indicated. 5. Why was such a short research period used? The problem arises was the matrix in the model balanced? and what are the consequences? 6. The analysis of the literature lacked a bit of discussion and demonstration of contradictions in this area.
Round 2
Reviewer 2 Report
Compared with the first draft, this manuscript has been improved, but there are still the following problems:
1. Figure 1. is not clear enough, please revise it
2. For example, in China, the amount of crop residues burned rose from about 58.46 billion tons to 68.21 billion tons between 2009 and 2019, accounting for 16.65% and 17.48% of the global share, respectively. What is the source of the data?
3. 3.2. Equation in Methodology without number
Author Response
Dear Reviewer,
Thank you very much for your affirmation and approval of my article.
1.Figure 1. is not clear enough, please revise it
Response: I have revise it with a high resolution image.
2. For example, in China, the amount of crop residues burned rose from about 58.46 billion tons to 68.21 billion tons between 2009 and 2019, accounting for 16.65% and 17.48% of the global share, respectively. What is the source of the data?
Response: This is come from the same source of next sentence. Moreover, I revise these two sentence to avoid repeat the same references.
3. 3.2. Equation in Methodology without number
Response: I have added the number